# Preparation of Cross-linked Chitosan Quaternary Ammonium Salt Hydrogel Films Loading Drug of Gentamicin Sulfate for Antibacterial Wound Dressing

**DOI:** 10.3390/md19090479

**Published:** 2021-08-25

**Authors:** Jingjing Zhang, Wenqiang Tan, Qing Li, Xiaorui Liu, Zhanyong Guo

**Affiliations:** 1Institute of Oceanology, Chinese Academy of Sciences, Qingdao 266071, China; jingjingzhang@yic.ac.cn; 2Key Laboratory of Coastal Biology and Bioresource Utilization, Yantai Institute of Coastal Zone Research, Chinese Academy of Sciences, Yantai 264003, China; wqtan@yic.ac.cn (W.T.); qli@yic.ac.cn (Q.L.); 3Center for Ocean Mega-Science, Chinese Academy of Sciences, 7 Nanhai Road, Qingdao 266071, China; 4College of Oceanography, Yantai University, Yantai 264005, China; sheri_xiaorui@163.com; 5University of Chinese Academy of Sciences, Beijing 100049, China

**Keywords:** antibacterial activity, chitosan quaternary ammonium salt, hydrogel films

## Abstract

Hydrogels, possessing high biocompatibility and adaptability to biological tissue, show great usability in medical applications. In this research, a series of novel cross-linked chitosan quaternary ammonium salt loading with gentamicin sulfate (CTMCSG) hydrogel films with different cross-linking degrees were successfully obtained by the reaction of chitosan quaternary ammonium salt (TMCS) and epichlorohydrin. Fourier transform infrared spectroscopy (FTIR), thermal analysis, and scanning electron microscope (SEM) were used to characterize the chemical structure and surface morphology of CTMCSG hydrogel films. The physicochemical property, gentamicin sulphate release behavior, cytotoxicity, and antibacterial activity of the CTMCSG against *Escherichia coli* and *Staphylococcus aureus* were determined. Experimental results demonstrated that CTMCSG hydrogel films exhibited good water stability, thermal stability, drug release capacity, as well as antibacterial property. The inhibition zone of CTMCSG hydrogel films against *Escherichia coli* and *Staphylococcus aureus* could be up to about 30 mm. Specifically, the increases in maximum decomposition temperature, mechanical property, water content, swelling degree, and a reduction in water vapor permeability of the hydrogel films were observed as the amount of the cross-linking agent increased. The results indicated that the CTMCSG-4 hydrogel film with an interesting physicochemical property, admirable antibacterial activity, and slight cytotoxicity showed the potential value as excellent antibacterial wound dressing.

## 1. Introduction

The skin is an important organ that maintains the osmotic balance of the body and protects the organism from environmental factors, and its integrity is very critical [1,2]. As the first protective barrier against a harsh environment, skin inevitably causes trauma. Although the skin has a certain self-healing ability, the side effects such as inflammation and secondary damage accompanying the long-term repair process make it of great demand in designing and developing clinical wound dressings [3,4]. Up to now, many different types of wound dressings have been designed, among which hydrogel-based wound dressings have attracted much attention [5,6,7].

Hydrogels are cross-linked three-dimensional networks comprised of hydrophilic polymer chains, which can absorb and retain large amounts of water or biological fluids [8,9]. Hydrogels, with large water uptake ability, have a microenvironment similar to the natural structure. Particularly, hydrogels have been regarded as notable candidates in wound dressings for drug delivery applications, owning to their good biocompatibility [10,11,12]. For example, Zhang et al. had successfully obtained the GMs/CMCS-OG composite hydrogels loaded with drugs. These composite hydrogels displayed fast gelation time, good stability, high mechanical properties, as well as strong antibacterial activity, which had potential for wide application in drug delivery and the anti-bacterial environment [8]. Hosseini et al. had synthesized hydrogel films based on basil seed mucilage and tested their thermal, physicochemical, morphological, rheological, mechanical properties; release effect on Tetracycline hydrochloride; and cytotoxicity. The results showed that the optimized hydrogel film could be used for drug delivery of wound dressing [13]. Thus, hydrogel wound dressing is an ideal choice for wound treatment.

Chitosan is a natural biopolymer derived from the *N*-deacetylation of chitin, found on crustaceans’ shells, an insect’s exoskeleton, fungi cell walls, etc. Due to the exceptional biocompatibility, readily biodegradability, and the unique biological activity, chitosan has attracted sufficient attention in numerous biomedical applications [14,15,16]. Nevertheless, the production of cross-linked chitosan in neutral and alkaline conditions is limited for the poor solubility of chitosan. Compared with chitosan, chitosan quaternary ammonium salt (TMCS) has higher water solubility and biological activity due to its quaternary ammonium salt group [17]. Moreover, because TMCS can dissolve under physiological conditions, it is widely used as an antimicrobial agent, flocculant, drug carrier, and tissue engineering material [18,19,20,21]. For instance, etofylline encapsulated mannose-anchored *N*,*N*,*N*-trimethyl chitosan nanoparticles were fabricated by Pardeshi et al. and their high loading and encapsulation of the etofylline, sustained drug release, lung tissue biocompatibility, stability, excellent particle dispersion, and aerodynamic behavior indicate the effectiveness of a prepared nanoformulation for pulmonary administration [22]. Thus, TMCS is one of the most promising candidates for preparing gentamicin sulfate release hydrogel films.

Gentamicin sulphate is an aminoglycoside broad-spectrum antibiotic, which can be used against serious infection especially caused by Gram-negative bacteria strains [23,24]. The mechanism of action is preventing the synthesis of bacterial protein by electrostatic binding with negatively charged phospholipids’ head groups. Subsequently, Gentamicin binds to specific ribosomal proteins leading to the production of nonfunctional complexes that result in mRNA misreading [25,26]. Furthermore, because the bacterial resistance of gentamicin sulphate is lower than other aminoglycosides, gentamicin sulphate has been widely used in the treatment of suppurative peritonitis, respiratory infections, intracranial infections, biliary infections, urinary tract infections, and infected lacerations [27,28].

In the present work, the cross-linked chitosan quaternary ammonium salt loading with gentamicin sulfate hydrogel films with different cross-linking degrees was prepared by the reaction of epichlorohydrin and TMCS under alkaline conditions, which could be used for antibacterial wound dressings. Moreover, chitosan quaternary ammonium salt film was also prepared for comparison. The obtained films were characterized using fourier transform infrared spectroscopy (FTIR), thermal analysis, and scanning electron microscope (SEM). Their various physicochemical properties were evaluated using water solubility, swelling degree, water vapor permeability, and mechanical behavior. We also assayed the gentamicin sulphate release ability of these novel hydrogel films at pH = 7.4 in vitro. The antibacterial activities against *Escherichia coli* (*E. coli*) and *Staphylococcus aureus* (*S. aureus*) were investigated using the zone of inhibition method. Additionally, L929 cells were used to carry out a cytotoxicity test of TMCS film and CTMCSG hydrogel films by CCK-8 assay. This work is aimed at preparing hydrogel films with good sustained-release effect, excellent antibacterial activity, and low cytotoxicity, so as to provide a new type of wound dressing with desirable performance. The preparation of TMCS-based hydrogel films not only shows great application potential in the application of wound dressings, but also involves the resource utilization of natural biomacromolecules. Therefore, this study will provide a theoretical basis for the development of chitosan in the field of medical materials.

## 2. Results and Discussion

### 2.1. Characterization of Films

The detailed synthetic strategy of chitosan hydrogel films was shown in Figure 1. Firstly, *N*,*N*,*N*-trimethyl chitosan (TMCS) was synthesized according to a previous method [17]. Then, the cross-linked chitosan quaternary ammonium salt (CTMCS) was synthesized by the reaction of TMCS and epichlorohydrin. Finally, gentamicin sulfate was loaded into CTMCS, and the cross-linked chitosan quaternary ammonium salt hydrogel films loading with gentamicin sulfate were obtained after drying. The structures of all products were confirmed by FTIR (Figure 1), thermal analysis (Figure 2), and SEM (Figure 3). Their mechanical properties, water content, water solubility, swelling degree, and water vapor permeability were also analyzed, and the results are shown in Table 1 and Table 2.

#### 2.1.1. FTIR Spectra

The FTIR spectra of chitosan quaternary ammonium salt (TMCS) and cross-linked chitosan quaternary ammonium salt with different cross-linking degrees (CTMCS) are shown in Figure 1. In all FT-IR spectra, the broad band around 3421 cm^−1^ was associated with the stretching vibrations of amine bonds (N–H) overlapped to the axial stretching of hydroxyl groups (O–H), and the peaks observed at 2919 cm^−1^ as well as 2881 cm^−1^ could be attributed to the stretching vibration of -CH as well as -CH_2_ in the chitosan chain. In addition, the band that appeared at 1654 cm^−1^ represented C = O stretching of the acetamide groups of chitosan [29]. The band at 1473 cm^−1^ was assigned to N–CH_3_ bending of the trimethyl group [17]. Compared with the FT-IR spectra of TMCS, the peaks at 1378 cm^−1^ (the bending vibration of -OH) and 1039 cm^−1^ (the stretching vibration of C-O-C) in the FT-IR spectra of CTMCS were significantly stronger. The results showed that the primary alcohol group of TMCS reacted with epichlorohydrin to produce additional secondary alcohol and aliphatic ether [30]. Furthermore, the absorption peak strength of C-O-C increased with the increase in the amount of epichlorohydrin, which indicated that the increased cross-linking agent was beneficial for the cross-linking reaction [17]. However, the peak intensity of the spectra of CTMCS-5 was weakened, which indicated that the amount of cross-linking agent was excessive. Taken together, the data confirmed that the epichlorohydrin was successfully cross-linked with TMCS.

#### 2.1.2. Thermal Analysis

The thermal stability of chitosan quaternary ammonium salt (TMCS) film and cross-linked chitosan quaternary ammonium salt loading with gentamicin sulfate (CTMCSG) hydrogel films was evaluated by thermogravimetric analysis. The resulting TGA curves and the first derivative of thermogravimetric (DTG curves) are given in Figure 2. Two weight-loss stages of thermal decomposition in the range of 40–500 °C were recorded in the thermograms, and similar trends were observed for all films. In the thermogram of TMCS film, two weight-loss stages occurred at the range of 40–110 °C and 130–300 °C, respectively. The initial weight-loss stage was corresponded to the release of crystal water. The major decomposition mainly occurred in the second stage, which was related to the thermal degradation of the TMCS backbone [31]. In the thermograms of CTMCSG, weight losses caused by the evaporation of crystal water and the degradation of the cross-linking sites were observed at around 40–140 °C. The subsequent thermal decomposition occurred at about 150–310 °C, which was associated with the thermal degradation of the CTMCSG backbone. The maximum decomposition temperatures of the cross-linked hydrogel films were observed at around 253–258 °C, which were evidently above that of TMCS film at 244 °C. This is because the covalent bond between the molecules of chitosan quaternary ammonium salt makes the structure of the hydrogel films more compact, thus improving its thermal decomposition performance [17]. Additionally, it is noteworthy that the temperatures at the maximum decomposition rate of CTMCSG films were proportional to the amount of cross-linking agent. The maximum decomposition temperatures of CTMCSG-1, CTMCSG-2, CTMCSG-3, CTMCSG-4, and CTMCSG-5 were 253.12 °C, 253.54 °C, 256.29 °C, 257.91 °C, and 255.45 °C, respectively. However, the maximum decomposition temperature of CTMCSG-5 was lower than that of CTMCSG-4 due to the excessive cross-linking agent, which was consistent with the result of FT-IR spectra. The results of thermal analysis showed that CTMCSG-4 film had the best thermal stability.

#### 2.1.3. Scanning Electron Microscopy 

The external photographs (top-right corner) and SEM surface images of TMCS film and CTMCSG films are depicted in Figure 3. The TMCS film was yellowish, transparent, and bright, while CTMCSG films with different cross-linking degrees were colorless, transparent, and frosted, as could be observed in their external photographs. The film microstructure is influenced by the arrangement of molecular structures during the initial dispersion process, and the film microstructure poses a critical influence on analyzing the physical and chemical properties of these films. As shown in the SEM image of uncross-linked TMCS film, due to the poor stability of chitosan quaternary ammonium salt in the environment, the surface of the membrane became very rough. Compared with TMCS film, the CTMCSG films presented regular textured surfaces. The CTMCSG films presented a relatively homogenous structure without any cracks or pores. This is due to the cross-linking reaction between the molecules of chitosan quaternary ammonium salt which results in more orderly cross-linking networks in the hydrogel films. Moreover, the SEM surface images of different hydrogel films also showed different morphological characteristics. Obviously, CTMCSG-4 film had the smoothest surface. The reason can be explained as follows: the CTMCSG-4 film has the highest degree of cross-linking, which is profit for keeping the TMCS macromolecular chain tightly arranged; thus, the structure of CTMCSG-4 film is the most compact among the five cross-linked hydrogel films.

#### 2.1.4. Thickness, Density, and Mechanical Properties

Table 1 summarizes some physicochemical properties of TMCS film and CTMCSG hydrogel films, including thickness, density, and mechanical properties. The thickness of TMCS film was measured as 93 ± 4 μm, and that of the CTMCSG hydrogel films was about 124 μm. Nevertheless, the densities of TMCS and CTMCSG hydrogel films were comparable. This is mainly due to the formation of the three-dimensional porous structure of hydrogels. Moreover, an increase in density was observed for the CTMCSG hydrogel films as the amount of cross-linking agent increased. The increase in density of hydrogel films might be likely due to the compact internal structure. In general, the density of CTMCSG hydrogel film was positively correlated with the amount of cross-linking agent and the compactness of film structure. In order to conform to the softness of skin, hydrogel films used as a wound dressing should possess appropriate mechanical properties, including tensile strength (TS) and elongation at break (EB). By comparison, it was found that the tensile strength of CTMCSG hydrogel films was lower than that of TMCS film, whereas the elongation at break was stronger than that of TMCS film. The results were probably attributed to the cross-linked network of CTMCSG hydrogel films. Although the formation of the cross-linking bond reduced the tensile strength of the hydrogel films, it significantly increased the elongation at break. This indicated that the hydrogel materials had excellent ductility. Specifically, when the amount of cross-linking agent was 4 mL, the tensile strength and elongation at break of the CTMCSG-4 hydrogel film were 7.44 MPa and 87.87%, respectively.

#### 2.1.5. Water Content, Water Solubility, Swelling Degree, and Water Vapor Permeability

Creating a moist environment for wound healing is essential to reduce the risk of surrounding skin complications. On this basis, the wound dressing needs to be able to absorb and retain the exudate while maintaining some moisture on the surface of the wound [32]. In an in vitro evaluation, because PBS can simulate the osmolarity and ion concentration of human body fluids, PBS was selected for swelling experiments in this study. The water content and swelling degree of TMCS film and CTMCSG hydrogel films are shown in Table 2. In gravimetric measurements, due to the presence of some surface water that cannot be completely removed, all values have high standard deviations. The water content of TMCS film was 19.58%, but the water content of the cross-linked hydrogel films increased to about 26%, which was mainly attributed to the water retention property of the hydrogels. In addition, because of the good water absorption, all CTMCSG hydrogel films possessed a high swelling degree. Moreover, their swelling degree decreased with the increasing in the degree of cross-linking, because the polymer network structure became stiff and compact, which made it difficult to stretch. It can be seen from Table 2 that the occurrence of the cross-linking reaction reduced the water vapor permeability of the prepared films. Furthermore, the water vapor permeability of hydrogel films was also affected by the degree of cross-linking. The increase in cross-linking degree enhanced the water vapor barrier properties of the hydrogel films with the lowered WVP values. This might be because the denser polymer structure led to greater resistance to the diffusion of the water molecules [33]. Indeed, all the CTMCSG hydrogel films showed good fluid absorption and retention characteristics, and could prevent excessive dehydration, making them suitable as wound dressings.

### 2.2. In Vitro Gentamicin Sulfate Release

Hydrogels are highly hydrated networks fabricated from a hydrophilic substance. Hydrogel materials play a very promising role in the field of controlled drug delivery because of its excellent biocompatibility, large porosity, high water content, strong swelling capacity, and high permeability to small hydrophilic molecules. Meanwhile, the drug release effect is one of the important indexes to evaluate the application of chitosan hydrogel film as medical wound dressing [34]. An excellent drug release effect can be considered if the drug release percentage reaches more than 50% and has obvious sustained release effect. Since TMCS film is easily soluble in water and does not obtain the ability of drug sustained release, Figure 4 only shows the gentamicin sulfate release effect of CTMCSG hydrogel films. As shown in Figure 4, the drug release rates of all hydrogel films were very fast at the beginning, and about 50% of gentamycin sulfate was released in the solution after 10 h. The explosive release might be attributed to the rapid diffusion of gentamycin sulfate from the surface region of hydrogel films. Then, owing to the cross-linked steric hindrance and long pathways, diffusion from the inner side occurred very slowly and the drug release was gradually increasing without burst release. Namely, the drug cumulative release rate reduced gradually and reached equilibrium at about 24 h. In addition, the gentamycin sulfate release rates somewhat varied among the five CTMCSG hydrogel films, which might be due to the varied cross-linking degree of epichlorohydrin with TMCS in hydrogels. To some extent, the increased cross-linking degree could slow down the drug release [34]. Therefore, CTMCSG-1 with the lowest cross-linking degree had the fastest drug release rate and drug release amount, while CTMCSG-4 possessed the relatively low gentamycin sulfate release rate and quantity. This conclusion is consistent with the results of hydrogel films characterization. Nevertheless, there was little difference in the gentamycin sulfate amount released from the five CTMCSG hydrogel films. For example, when the release time was 48 h, the cumulative release rates of CTMCSG-1 hydrogel film and CTMCSG-4 hydrogel film were 64.15% and 59.60%, respectively. In conclusion, the prepared hydrogel films show linear drug release characteristics, indicating the sustained drug release behavior. This could be considered as a good indicator for their future applications in the area of wound dressings. Moreover, the cumulative drug release property makes the prepared CTMCSG hydrogel films a promising candidate in the field of medical materials.

### 2.3. Antibacterial Evaluation

The ideal biomaterial medical dressings should possess an excellent antibacterial property to prevent bacterial infections at wound bed and reduce the episode of inflammatory reactions, so as to promote wound healing. In this study, *Staphylococcus aureus* (Gram-positive bacteria) and *Escherichia coli* (Gram-negative bacteria), both of which are prevalent bacteria in wound infections, were selected to investigate the antibacterial property of chitosan quaternary ammonium salt (TMCS) film and cross-linked chitosan quaternary ammonium salt loading with gentamicin sulfate (CTMCSG) hydrogel films via the zone of inhibition method [7]. The antibacterial effects were evaluated by measuring the inhibition zone diameter, and the larger diameter represented the stronger antibacterial activity. As presented in Figure 5, the inhibition zone could be observed obviously in CTMCSG hydrogel films, whereas the TMCS film showed almost no inhibition zone. This result indicated that CTMCSG hydrogel films could effectively inhibit the bacterial proliferation around them. For example, CTMCSG-1, CTMCSG-2, CTMCSG-3, CTMCSG-4, and CTMCSG-5 hydrogel films had inhibition zone diameters of 31.29, 30.35, 30.37, 36.37, and 30.07 mm against *E. coli*, respectively. Thereinto, CTMCSG-4 possessed the largest inhibitory diameter and exhibited the best inhibitory ability of *E. coli*. There was little difference in the antibacterial diameters of the samples against *S. aureus*, which were 28.19, 26.33, 32.97, 33.14, and 28.53 mm, respectively. Indeed, chitosan-based hydrogel material showed a certain antibacterial manner. Moreover, gentamicin sulphate was proved to exhibit great in vivo antibacterial efficacy and could significantly promote the healing of purulent wounds that were infected by *E. coli* and *S. aureus*. Thus, the synergistic effect of gentamicin sulphate and cross-linked chitosan quaternary ammonium salt endowed the augmentation of antibacterial activities of CTMCSG hydrogel films, and they could be considered as a potential antibacterial wound dressing. The conclusion has further shown the great potential of CTMCSG hydrogel films in the field of medical materials.

### 2.4. Cytotoxicity Analysis

Cytotoxicity is an important evaluation criterion for medical dressing hydrogel films as they provide the environment for cell growth. In the present study, the results of the viability of L929 cells cultured after 24 h, 48 h, and 72 h in the presence of CTMCSG hydrogel film extracts are presented in Figure 6. The experiment was carried out using a CCK-8 test. The normal cell morphology is spindle or oval. On the contrary, when the cell is round with dead cell fragments, it indicates low cell activity. The cytotoxicity of all samples decreased with time. For TMCS film, relatively high cell viability could be observed, indicating chitosan quaternary ammonium salt had a low cytotoxicity for the L929 cells. For CTMCSG-1 film, when cultured for 24 h, the cell survival rate was 73.41%. When the culture time increased to 48 h, the cell survival rate reached to 110%, indicating that hydrogel material was beneficial to cell proliferation. The viability values of cells treated with CTMCSG-2, CTMCSG-3, CTMCSG-4, and CTMCSG-5, by contrast, were lower, and their cell viabilities at a culture time of 24 h were 49.60, 49.14, 61.22, and 51.59%, respectively. When the culture time was 72 h, the survival rates of L929 cells cultured with CTMCSG-2, CTMCSG-3, CTMCSG-4, and CTMCSG-5 extracts were over 60%. For CTMCSG-4 and CTMCSG-5 hydrogel films, their cell viabilities were 81.94 and 93.74%, which indicated that these hydrogel films were fabricated with low cytotoxicity to the L929 cells. Therefore, the combination of the comprehensive evaluations of the structural characteristic, swelling degree, water vapor permeability, mechanical property, and biological activity of all CTMCSG hydrogel films revealed that the CTMCSG-4 film with excellent antibacterial activity and slight cytotoxicity could be used as a potential medical wound dressing.

## 3. Materials and Methods 

### 3.1. Materials

Chitosan, for which the degree of deacetylation is 83% and the average molecular weight is 200 KDa, was purchased from Qingdao Baicheng Biochemistry Corp. (Qingdao, China). *N*-methyl-2-pyrrolidone (NMP), ethanol, sodium hydroxide, epichlorohydrin, glycerin, and iodomethane were purchased from Sinopharm Chemical Reagent Co., Ltd. (Shanghai, China). Sodium iodide and gentamicin sulfate were purchased from Sigma-Aldrich Chemical Corp. (Shanghai, China). All reagents were used as received without further purification, and the detailed information is shown in Table 3.

### 3.2. Preparation of Cross-linked Chitosan Quaternary Ammonium Salt Hydrogel Films Loading with Gentamicin Sulfate (CTMCSG)

Firstly, chitosan quaternary ammonium salt (TMCS) was obtained, and the synthesis process was followed by the previous methods [35]. Then, 2.0 g of TMCS was uniformly dissolved in 100 mL of deionized water at 70 °C. Subsequently, a certain amount of epichlorohydrin (1, 2, 3, 4, and 5 mL, respectively) was added while stirring at 70 °C for the cross-linking reaction of TMCS for 4 h. During this period, the pH value of the reaction mixture was maintained at around 10 by adding 5% sodium hydroxide solution. After the cross-linking reaction, the pH value of the mixture was adjusted to 7 with 20% solutions of hydrochloric acid, and a colorless cross-linked chitosan quaternary ammonium salt gel solution was obtained. The reaction mixture was dialyzed with distilled water for 2 days. Then, under continuous stirring, 0.2 mL of glycerin and 0.1 g of gentamicin sulfate were added into the purified gel solution. Finally, the gel solution mixture was dried at 30 °C for 24 h to obtain cross-linked chitosan quaternary ammonium salt loading with gentamicin sulfate (CTMCSG) hydrogel films. The dosage of epichlorohydrin for the preparation of CTMCSG-1, CTMCSG-2, CTMCSG-3, CTMCSG-4, and CTMCSG-5 was 1, 2, 3, 4, and 5 mL, respectively. The synthesis route of CTMCSG is shown in Figure 1. As a reference, chitosan quaternary ammonium salt film (TMCS) was also prepared according to the above operation, except with no cross-linking reaction.

### 3.3. Characterization of Films

#### 3.3.1. Fourier Transform Infrared (FT-IR) Spectroscopy

The infrared spectra of chitosan quaternary ammonium salt and cross-linked chitosan quaternary ammonium salt were obtained using Nicolet iS50 Fourier Transform Infrared Spectrometer (Thermo, Waltham, MA, USA) with a transmission mode covering the frequency ranging from 4000 to 400 cm^−1^ at 25 °C. All the tested samples were scanned 32 times using the KBr pellet method (weight ratio 1:100).

#### 3.3.2. Thermal Analysis

The thermal stability of chitosan quaternary ammonium salt film and cross-linked chitosan quaternary ammonium salt loading with gentamicin sulfate hydrogel films was analyzed by a thermogravimetric analyzer (Mettler 5MP, Mettler-Toledo, Greifensee, Switzerland). The film sample was heated from 40 °C to 500 °C at a constant rate of 10 °C·min^−1^ under continuous nitrogen flow of 20 mL·min^−1^ to obtain derivative thermogravimetric (DTG) data and thermogravimetric analysis (TGA).

#### 3.3.3. Scanning Electron Morphology

The morphology of samples was characterized using scanning electron microscopy (SEM, S-4800, Hitachi, Japan). To be brief, the dried films were put on a sample-holder for the observation by SEM.

#### 3.3.4. Thickness and Density

A digital micrometer (Jingcheng, China) with 0.001 mm precision was used to measure the thickness of films at five randomly selected test points. The average thickness value of the film was used for subsequent volume calculations. The film density was calculated from the ratio of weight to volume of the film.

#### 3.3.5. Mechanical Properties

The mechanical property of films, including tensile strength (TS) and elongation at break (EB), was recorded by a universal tensile tester (Instron 5848 MicroTester, High Wycombe, UK). The rectangular strips of films were mounted in a tensile grip and operated with a 100 N load cell at a cross-head speed of 1 cm/min until breakage. The average TS (MPa) and EB (%) were determined from the resulting stress-strain curves. The mechanical measurements of each film were repeated three times, and the average values were used for each hydrogel film.

#### 3.3.6. Water Content and Swelling Degree

Firstly, the films (3 cm × 1 cm) were weighed (W_0_) in pre-weighed glass Petri dishes. The film sample was subsequently dried at 75 °C for 24 h in a hot oven, and the initial constant dry weight (W_1_) was measured. Then, the dried film was immersed in 30 mL of phosphate buffered saline (PBS) solution (pH 7.4) for 24 h at 37 °C, and the wet hydrogel film was weighed after blotting the excess surface solution with filter paper (W_2_). Measurement of each film was repeated three times. The water content and swelling degree of sample films were calculated using Equations (1) and (2), respectively:(1)$$\mathrm{Water}\;\mathrm{content}\;\left(\%\right)=\left(W_{0}-W_{1}\right)/W_{0}\times 100\;$$
(2)$$\;\mathrm{Swelling}\;\mathrm{degree}\;\left(\%\right)=\frac{W_{2}-W_{1}}{W_{1}}\times 100$$

#### 3.3.7. Water Vapor Permeability

The water vapor permeability (WVP) of the hydrogel films was recorded gravimetrically according to the dry cup method of ASTM E96-95 with minor modifications [36]. The hydrogel film (4 cm in diameter) was sealed on the top of a circular test cup filled with 18 mL of distilled water. The cup was measured to get initial weight and stored in the closed desiccator filled with anhydrous silica gel at 25 °C. In the testing process, the partial pressure difference between the two sides of the hydrogel film created a driving force for water vapor to be transported through the film, which caused a decrease in the weight of the test cup. The weight loss of each cup was recorded using an analytical balance every 2 h for a continuous 24 h and plotted as a function of time. The WVP (g·m/m^2^·Pa·s) of the hydrogel films was calculated using Equation (3):(3)WVP(%)=(Δm×x)/(s×Δp×t)
where Δm is the weight loss (g); x is the mean thickness of film (m); s is the permeation area of the films (m^2^); ΔP is the partial water vapor pressure difference (Pa) between the inner and outer film surfaces at 25 °C; and *t* is the test interval (s).

### 3.4. Evaluation of Sustained Release Performance

The sustained release performance of hydrogel films loading with gentamicin sulfate was investigated using the RC1207DP dissolution tester (Tianjin, China) and ultraviolet spectrophotometer (UV spectrophotometer, T6, Pgeneral, Beijing, China). Firstly, 200 mL of phosphate buffered salines (pH = 7.4) were utilized as the dissolution media to ensure the sink condition. Then, the hydrogel films were immersed in phosphate buffered salines and were left in a shaking water bath at 37 °C. Finally, samples were fetched according to predetermined intervals, and the same volume of phosphate buffered salines was compensated immediately. The release of gentamicin sulphate was estimated by using an ultraviolet spectrophotometer with the measuring absorption at 246 nm [37,38].

### 3.5. Antibacterial Assay

The antibacterial ability of chitosan quaternary ammonium salt film and cross-linked chitosan quaternary ammonium salt loading with gentamicin sulfate hydrogel films was determined using the zone of inhibition method [39]. Firstly, the autoclaved bacterial nutrient agar media was prepared. Then, the cultured media was poured into sterile plates. After the media was solidified, 0.1 mL *E. coli* and *S. aureus* (10^5^ CFU/mL) were inoculated and evenly coated with a sterile spatula. Next, film with a diameter of 5 mm was placed in the center of each plate. Finally, the bacterial plates were incubated at 37 °C overnight, and zones of inhibition were measured after 24 h. Generally, a greater inhibition zone indicated higher antibacterial activity.

### 3.6. Cytotoxicity Assay

Cytotoxicity of hydrogel films on L929 cells was determined by CCK-8 assay in vitro. Firstly, the extract solution of the CTMCSG hydrogel films was obtained by immersing the materials in serum cell culture medium (DMEM medium with 1% mixture of penicillin and streptomycin and 10% fetal calf serum) for 24 h (at 37 °C), and the obtained extract solution was stored at 4 °C. Then, L929 cells were seeded on 96-well flat-bottom culture plates with the concentration of 1.0 × 10^5^ cells in 100 µL of RPMI medium and incubated at 37 °C under CO_2_ atmosphere (5%). After 24 h of cell attachment, the DMEM medium was removed, and cells were cultured with CTMCSG hydrogel film extracts for 24 h, 48 h, and 72 h. After incubation, 10 µL of CCK-8 solution were dropped into each well, and the plates were incubated for more 4 h. The absorbance of every well was recorded using a microplate reader (DNM-9602G, Thermo Multiskan Ascent, Waltham, MA, USA) at 450 nm. The cytotoxicities were calculated by the following formula:(4)$$\mathrm{Cell}\;\mathrm{viability}\;\left(\%\right)=\frac{A_{\mathrm{sample}}-A_{\mathrm{blank}}}{A_{\mathrm{negative}}-A_{\mathrm{blank}}}\times 100$$
where A_sample_ is the absorbance of the samples (containing cells, CCK-8 solution, and sample solution); A_blank_ is the absorbance of the blank (containing RPMI medium and CCK-8 solution); and A_negative_ is the absorbance of the negative (containing cells and CCK-8 solution).

### 3.7. Statistical Analysis

Three times for every experiment were repeated, and the data were reported as mean ± standard deviation (SD), *n* = 3. Significant difference analysis was determined according to Scheffe’s multiple range test. The level of *p* < 0.05 was defined as statistically significant.

## 4. Conclusions

Novel CTMCSG hydrogel films with different cross-linking degrees were successfully prepared via the reaction of TMCS and epichlorohydrin. The characterization of films was confirmed by means of FT-IR, thermal analysis, and SEM. The results of structural characterization showed that the amount of cross-linking agent had a certain influence on the physicochemical property, and CTMCSG-4 hydrogel film had the highest degree of cross-linking as well as the densest structure. Moreover, the denser structure increased the thermal stability, mechanical property, whitish index, water content, swelling degree, and a reduction in water vapor permeability of the hydrogel films. Meanwhile, the gentamicin sulfate release of novel hydrogel films was assayed in vitro, and the result indicated that the novel hydrogel films possessed good drug release performance in the release media, with more than 50% of the drug released in 24 h. Additionally, the loading of gentamicin sulfate in hydrogels remarkably enhanced its antibacterial property against both *E. coli* and *S. aureus*. The CCK-8 study also confirmed the cytocompatibility of the developed CTMCSG hydrogel films. The above results indicated that the combination of chitosan quaternary ammonium salt and gentamicin sulfate resulted in low-cytotoxic hydrogel films with an excellent physicochemical property and antibacterial activity, which could be well applied in the field of wound dressing. From the perspective of promise, the prepared CTMCSG hydrogel films have significant potential as medical materials.

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
