# Peer review of "Preparation of Cross-linked Chitosan Quaternary Ammonium Salt Hydrogel Films Loading Drug of Gentamicin Sulfate for Antibacterial Wound Dressing"

_marinedrugs, 2021, doi:10.3390/md19090479_

Round 1
Reviewer 1 Report
One of the aims of this research is "to provide a theoretical basis for the development of chitosan in the field of medical materials" (page 2, line 99) and the incremental development of chitosan-based films with well known antibacterial properties.
But the theoretical basis has not been duly discussed in detail in the text of manuscript and in Conclusions.
The Introduction presented appropriate and sufficient description of current state-of-the-art in the development of chitosan-based films to be used as drug delivery system in wound dressings.
The experimental methods are adequately described and results are clearly presented.
Author Response
Dear reviewer,
Thank you for your comments concerning our manuscript entitled “Preparation of cross-linked chitosan quaternary ammonium salt hydrogel films loading drug of gentamicin sulfate for antibacterial wound dressing”. Those comments are all valuable and very helpful for revising and improving our paper. We have studied comments carefully and have made corrections which we hope meet with approval. The main corrections in the manuscript are as following:
- One of the aims of this research is "to provide a theoretical basis for the development of chitosan in the field of medical materials" (page 2, line 99) and the incremental development of chitosan-based films with well known antibacterial properties.
But the theoretical basis has not been duly discussed in detail in the text of manuscript and in Conclusions.
Answer: Thank you for your kind suggestions and according to your recommendation, we have discussed the practical applications of CTMCSG hydrogel films in the field of medical materials combined with the experimental results in the text of manuscript and in Conclusions (Lines 249-251, Lines 276-277, Lines 435-439).
- The Introduction presented appropriate and sufficient description of current state-of-the-art in the development of chitosan-based films to be used as drug delivery system in wound dressings.
The experimental methods are adequately described and results are clearly presented.
Answer: Thank you for your comments.We'll continue our research work in this area.
The revised manuscript has been submitted to journal. We hope that the responses and the revised manuscript adequately address your concerns. Thank you for your time and concerns.

Reviewer 2 Report
The paper is interesting and the obtained results are useful in advancing the knowledge in the field of chitosan related materials, but the writing needs serious improvement before publishing. Bellow you can find some observations, but please keep in mind that the entire manuscript must be revised from the point of view of English language.
- The definition of hydrogels is not the most appropriate. Line 43 must be rephrased. The authors are kindly advised to use hydrogels instead of hydrogel, the plural form being more general.
- Line 64 – replace stronger with higher.
Line 65- replace in with as
- Scheme 1- Crosslinked reaction – must be changed in crosslinking reaction. Moreover, due to the fact that the scheme contains also preparation details, not only the synthetic pathway used for chitosan’s derivatives, the caption must be rephrased.
- Line 106 – an appropriate reference must be cited.
- Line 122 – was belonged is not appropriate.
- The authors are advised to insert a table with the samples’ codes and the used amount of reagents in order to be clearer for the readers.
- Caption 2.1.3 Must be replaced with scanning electron microscopy or Morphology by scanning electron microscopy
- Line 172-174 There are some misinterpretations in the morphology part. There are no particles – maybe the sample was not completely dissolved for film preparation. If there are particles, an explanation must be found.
- Line 176-178 please rewrite for clarity.
- Line 228 is too general. The use of hydrogels in drug delivery applications is a consequence of their intrinsic properties such as porosity, high water content, swelling ability and not of their “structural properties”.
- Line 287-289 Please move to the experimental part.
Author Response
Dear reviewer,
Thank you for your comments concerning our manuscript entitled “Preparation of cross-linked chitosan quaternary ammonium salt hydrogel films loading drug of gentamicin sulfate for antibacterial wound dressing”. Those comments are all valuable and very helpful for revising and improving our paper. We have studied comments carefully and have made corrections which we hope meet with approval. The main corrections in the manuscript are as following:
- The definition of hydrogels is not the most appropriate. Line 43 must be rephrased. The authors are kindly advised to use hydrogels instead of hydrogel, the plural form being more general.
Answer: Thank you for your kind suggestions and according to your recommendation, we have rewritten this sentence (Lines 45-46) and used hydrogels instead of hydrogel in the text of manuscript.
- Line 64 – replace stronger with higher. Line 65- replace in with as.
Answer: Thank you for your kind suggestions and according to your recommendation we have revised them.
- Scheme 1- Crosslinked reaction – must be changed in crosslinking reaction. Moreover, due to the fact that the scheme contains also preparation details, not only the synthetic pathway used for chitosan’s derivatives, the caption must be rephrased.
Answer: Thank you for your kind suggestions. We have changed “Cross-linked reaction” in “Cross-linking reaction” in Scheme 1 and have carefully checked the accuracy of the writing about “Cross-linking reaction” in whole text. By reviewing the literature, we found that most of the studies used “Cross-linked products” rather than “Cross-linking products”, so, we did not modify the “cross-linked chitosan quaternary ammonium salt hydrogel films”.
- Line 106 – an appropriate reference must be cited.
Answer: Thank you for your kind suggestions and according to your recommendation we have added the reference.
- Line 122 – was belonged is not appropriate.
Answer: Thank you for your kind suggestions. We have revised it to “was assigned to”.
- The authors are advised to insert a table with the samples’ codes and the used amount of reagents in order to be clearer for the readers.
Answer: Thank you for your kind suggestions and according to your recommendation we have inserted a table with the reagents’ product code and specification in Section 3.1.
- Caption 2.1.3 Must be replaced with scanning electron microscopy or Morphology by scanning electron microscopy.
Answer: Thank you for your kind suggestions and according to your recommendation we have replaced it.
- Line 172-174 There are some misinterpretations in the morphology part. There are no particles – maybe the sample was not completely dissolved for film preparation. If there are particles, an explanation must be found.
Answer: Thank you for your kind suggestions. The surface image of uncross-linked TMCS film is very rough, which is caused by the poor stability of chitosan quaternary ammonia salt in the environment. The explanation is also reflected in the manuscript (Lines 165-167).
- Line 176-178 please rewrite for clarity.
Answer: Thank you for your kind suggestions and according to your recommendation we have rewritten these sentences (Lines 169-176).
- Line 228 is too general. The use of hydrogels in drug delivery applications is a consequence of their intrinsic properties such as porosity, high water content, swelling ability and not of their “structural properties”.
Answer: Thank you for your kind suggestions and according to your recommendation we have rewritten the sentence to “Hydrogels are highly hydrated networks fabricated from hydrophilic substance. Hydrogel materials play a very promising role in field of controlled drug delivery because of its excellent biocompatibility, large porosity, high water content, strong swelling capacity, and high permeability to small hydrophilic molecules. Meanwhile, drug release effect is one of the important indexes to evaluate the application of chitosan hydrogel film as medical wound dressing.” (Lines 225-229).
- Line 287-289 Please move to the experimental part.
Answer: Thank you for your kind suggestions. According to your recommendation we have rewritten the sentence to “In the present study, the results of viability of L929 cells cultured after 24 h, 48 h, and 72 h in the presence of CTMCSG hydrogel film extracts are presented in Fig. 6.” and have supplemented the corresponding experimental part (Section 3.6).
The revised manuscript has been submitted to journal. We hope that the responses and the revised manuscript adequately address your concerns. Thank you for your time and concerns.
